# Follow-Up of a Cohort of Patients with Post-Acute COVID-19 Syndrome in a Belgian Family Practice

**DOI:** 10.3390/v14092000

**Published:** 2022-09-09

**Authors:** Marc Jamoulle, Gisele Kazeneza-Mugisha, Ayoub Zayane

**Affiliations:** 1HEC Information Sciences, University of Liège, 4000 Liege, Belgium; 2Faculty of Medicine, University of Mons, 7000 Mons, Belgium; 3Centre Medical Janson, 6000 Charleroi, Belgium

**Keywords:** general practice, post-acute COVID-19 syndrome, medically unexplained symptoms, encephalopathy, narrative medicine, single photon emission computed tomography (SPEC CT scan), health status indicator

## Abstract

Fifty-five patients who suffered from COVID-19, who were still very ill after several months, with extreme fatigue, effort exhaustion, brain fog, anomia, memory disorder, anosmia, dysgeusia, and other multi-systemic health problems have been followed in a family practice setting between May 2021 and July 2022. Data extracted from the medical records of the 55 patients (40 women), mean age 42.4 (12 to 79 years), and a qualitative study of 6 of them using a semi-open-ended questionnaire allowed to highlight the clinical picture described by WHO as post-acute COVID-19 syndrome (PACS) also known as long COVID. We used brain single-photon emission computed tomography (SPECT-CT) in thirty-two patients with a high severity index and a highly impaired functional status, demonstrating vascular encephalopathy in twenty nine patients and supporting the hypothesis of a persistent cerebral vascular flow disorder in post COVID-19 condition. The patients will benefit from the consortium COVID Human Genetic Effort (covidhge.com) to explore the genetic and immunological basis of their problem, as 23/55 cases don’t have immunological certainty of a COVID-19 infection. There is no known verified treatment. Analyzing the data from the first 52 patients, three categories of patients emerged over time: 16 patients made a full recovery after 6–8 months, 15 patients were able to return to life and work after 12–18 months with some sequelae, both groups being considered cured. In the third group, 21 patients are still very ill and unable to resume their work and life after 18 months. The biopsychosocial consequences on patients’ lives are severe and family doctors are left out in the cold. It is necessary to test the reproducibility of this description, conducted on a small number of patients. Nevertheless, identifying, monitoring and supporting these patients is a necessity in family medicine.

## 1. Introduction

COVID-19 is no longer just an acute syndrome [1,2]. In nearly 20% to 35% of post acute COVID-19 patients [3,4] it can develop into a disabling health problem, sometimes lasting several months, called PACS by WHO, first called long COVID by the patients themselves and appeared under this name in the literature in the summer of 2020 [5,6]. PACS remains also very difficult to define [7]. A Delphi consensus, led by the WHO, was necessary to arrive at a still imprecise definition:


*PACS occurs in individuals with a history of probable or confirmed SARS-CoV-2 infection, usually 3 months from the onset of COVID-19 with symptoms and that last for at least 2 months and cannot be explained by an alternative diagnosis. Common symptoms include fatigue, shortness of breath, cognitive dysfunction but also others, and generally have an impact on everyday functioning. Symptoms may be new onset following initial recovery from an acute COVID-19 episode, or persist from the initial illness. Symptoms may also fluctuate or relapse over time [8].*


It is the first disease to be named by patients themselves through exchanges on social networks (see some internet website addresses in Appendix A) by patient advocacy groups whose members identify themselves as long COVID cases. An estimated 1.8 million people living in private households in the UK (2.4% of the population) were experiencing self-reported PACS in August 2022 [9]. These numerous testimonies and the considerable number of reported cases show that this debilitating disease is becoming a serious public health problem.

Among COVID-19 survivors more than 50% had one or more long COVID symptoms recorded during the 6-month period post infection [10]. Diagnosis of PACS remains difficult, as the syndrome encompasses distinct groups of heterogeneous symptoms [11,12,13]. These have poor diagnostic properties as they may overlap, evolve over time and are sometimes difficult to link to COVID-19. Patients with myalgic encephalomyelitis/chronic fatigue syndrome (ME/CFS) are known to have (highly) similar health complaints [14]. Symptoms affect multiple systems, and often occur after a relatively mild acute illness [15] or an influenza-like illness. They can be multiple and of varying intensity depending on the areas of the body affected. Unbearable fatigue, brain fog [16] and myalgia are the most common symptoms. Cognitive disorders, memory and attention deficits with very impaired quality of life [17], anomia, dysarthria, frontal behavioural disorders, autonomic dysregulation, headaches, dyspnoea, anosmia, dysgeusia, skin or digestive disorders, psychosocial distress, loneliness, anxiety, depression and sleep disorders have been associated with PACS [18,19,20,21,22]. In addition, post-COVID patients have an increased risk of psychotic disorders, dementia and epilepsy or seizures [23].

The absence of specific markers means that the diagnosis is based on the patient’s word, which is not without medico-legal consequences [24]. To date, we haven’t reached a full understanding of what PACS really is, nor its natural history [25]. A major UK biobank longitudinal study shows that changes in brain structure were found in some patients with SARS-Cov-2 after several months and in particular a greater reduction in grey matter thickness and a reduction in global brain size [26]. Its pathophysiology is still unclear [27] and there is no specific treatment [28]. Moreover, although some research seems encouraging, it is not clear that vaccination protects against PACS [3,29]. The long-term consequences of COVID-19 can be very devastating [30] while the socio-economic impact of long COVID on work and social life could be very important [31].

The family doctor can certainly provide information and support to these distressed and sometimes stigmatized patients [32].

Since May 2021, several long-standing patients (known by the doctor sometimes for years), come to the family practice unit with a set of similar unexplained symptoms. Neither the biological examinations nor the usual imaging techniques explained the condition of these patients. However, the patients presented a serious alteration of their physical and mental state. The family doctor is well positioned to explore a problem [33] which is then turned into a research question. It is about understanding how COVID-19 is responsible for unexpected symptoms which overwhelm the patients, their families and their employers, who may not understand the situation. It is not unusual for family doctors to be confronted with a case of MUS [34]. The certainty of the diagnosis itself is low and is primarily clinical. A careful history [35] will be essential to support the patient, to identify and name [36] this tiring PACS syndrome which seems endless and without remedy.

In an attempt at answering these questions, the data and testimonies of a cohort of patients seen in daily practice were collected, analyzed and discussed based on the literature as well as exchanges and discussions with colleagues. The aim of this study is thus to provide a qualitative and quantitative description of the health status of fifty-five patients (full data available for 52 of them) with PACS requiring care from May 2021 to July 2022 in a general practice in Charleroi, Belgium. This study aims to contribute to the description of PACS, the understanding of the suffering of the patients, and physio-pathological phenomena of long duration triggered by SARS-CoV-2.

## 2. Methods

Human subjects research, i.e., “*research involving the collection, storage, or use of private data or biological samples from living individuals*” [37] is typical to family medicine. Such research cannot always be planned in advance. The research topic emerges, imposes itself, even though the EMR collection of data has already started. This particular questioning leads to a review of the literature, which modifies the research protocol and the attitude towards the patients who enter the cohort. It is an action research [38] in which central concerns, improvement in practice, increased knowledge and understanding are linked together.

### 2.1. Clinical Data Collection

In May 2021, in a fee for service group practice of two family doctors, serving about 2500 patients in a deprived area in the city of Charleroi, Belgium, taking into account an increasing number of patients with unexplained symptoms after being sick with COVID-19, a data collection was initiated on thirty-two patients of the practice. After the publication of a first research report [39] and its dissemination on social networks, twenty-four patients from other treating physicians joined the cohort. In June 2022, a standardized follow-up form has been designed for patients with clinical signs of PACS met in the daily busy practice of the two practitioners (MJ & AZ). In addition to the biographical characteristics, this form lists the symptoms of acute COVID, the dates of PCR if available, the severity indicators, the examinations carried out, the evolution of the clinical state along the time and the paraclinical examinations requested. Comorbidity is recorded by the ICPC-2 [40]. These records are unidentified and shared with immunologists and geneticists for further studies (see Section 2.3).

Clinical certainty is considered here, i.e., the conviction by the practitioner that, as stated by Llewelyn, “*the anamnestic and paraclinical elements at his/her disposal allow him/her to reject with a high probability that the case presented is due to another condition*” [41]. Only cases deemed clinically demonstrative of PACS are included. Biological certainty refers here to the availability of a positive PCR test.

Severity is assessed by the doctor using the DUSOI/WONCA [42,43]. The index is based on four parameters: symptom status, complications, prognosis over the next six months without treatment, and treatability, i.e., the expected response to treatment. The score ranges from 0 (no severity) to 5 (extremely severe). The DUSOI/WONCA form with instructions for use is available online [44].

The functional status is assessed by the patients themselves using COOP/WONCA [45]. General condition, ability to perform daily activities, physical condition, emotional management, ability to have a social life and changes in health status are indicated on the 6 corresponding charts by patients rating themselves from 1 (good performance) to 5 (cannot do it). The overall index will have a value from 6 (excellent form) to 30 (totally impaired functional status). A manual in several languages and instructions for use is available online [45].

### 2.2. Data Collection from Narrative Medicine and Qualitative Approach

A detailed clinical medical case report was written by the doctor for each patients with highly significant signs of PACS. The first six patients were visited by GK, a medical student who was not involved in their care. According to the principles of narrative medicine [35], GK conducted the interviews on the basis of standardized semi-open questions. These recorded interviews were transcribed in full and will be also used for further qualitative research. GK then reviewed and corrected if necessary the clinical case reports with each patient.

### 2.3. Laboratory

None of the laboratory tests ordered usually in primary care proved to be contributory and consequently are not mentioned here. The results of PCR tests and COVID humoral serology are indicated with the dates of their performance when available. COVID-19 serology is not routinely requested in Belgium, as this comes at a charge.

All but four patients have accepted to provide a blood sample for genetics and immunological analysis by members of the network COVID Human Genetic Effort (Covidhge.com) (accessed on 1 September 2022), an international consortium aiming to discover the human genetic and immunological bases of the various clinical forms of SARS-CoV-2 infection and the particular characteristics of PACS patients [12,46]. These immunological tests should also remove diagnostic uncertainty for patients without a PCR test. Indeed, the uncertainty of the biological diagnosis can have psychological, medico-legal and clinical consequences. The results of these analyses are not yet available and will be published in the near future.

### 2.4. Nuclear Imaging

A Positron emission tomography with 18 fluoro-D-glucose integrated with computed tomography (18FDG PET-CT) can provide precise information such as the hypo-metabolism that affects certain brain areas in certain PACS with a strong neurological component [19,47,48,49,50,51], and is considered the gold standard of nuclear imaging.

Single Photon Emission Tomography-Computed Tomography (SPECT-CT), on the other hand, is more accessible and may reveal a metabolic brain disorder similar to that found in Alzheimer’s disease or stroke [52]. SPECT-CT 99mTc-ethyl cysteinate dimer (ECD Tc-99m), an old technique [53] coupling tomoscintigraphic and CT images acquired successively during the same examination, allows the assessment of brain perfusion using complex technetium-99m (Tc-99m) labeled molecules that are able to cross the blood-brain barrier [54]. The binding of these tracers is dependent on cerebral blood flow. Tc-99m is used for brain perfusion studies because of its high first-pass extraction fraction and high affinity for the brain [55]. With due precaution, Tc-99m can be disposed of as normal waste.

This examination, easily available in primary care in Belgium, was requested only in patients judged by the physician to be strongly or very strongly affected or with a severely impaired functional status. Brain CT scans and Nuclear Magnetic Resonance (NMR) scans, if any, were also collected.

### 2.5. Statistical Analysis

Binary comparisons were performed by aggregating patient groups 1 and 2 (cured) compared to group 3 (still ill)(see Table 1). Data were analyzed by using SPSS Statistics for Windows, Version 26.0 (IBM SPSS Statistics for Windows, Armonk, NY: IBM Corp). Descriptive data were presented as *n* and percentage or mean and standard deviation (SD). Categorical variables were compared using the chi-square or Fisher’s exact tests while the independent samples *t*-test or the Mann-Whitney U test was used to check for differences between the cured versus still ill patients regarding age, COOP total scores, and days since COVID symptoms. *p*-values below 0.05 were considered statistically significant (see Table 2).

### 2.6. Ethics

Patients whose medical records are managed by doctors in the patient-doctor Belgian contractual framework, have expressly given their written consent to the use and publication of their personal data in an anonymous manner. The ethics committee of the University Hospital of Liege, Belgium, gave its full approval to this study under the number 2022/23.

## 3. Results

### 3.1. Clinical Data

A total of fifty-five patients presenting with unusual or medically unexplained symptoms (MUS) were followed up in family medicine. practice between May 2021 and July 2022. The fifty-five patients, aged 12-79 years (M= 42.9 years at start of study), 40 women (72.7%), 15 men (27,3%), were clinically suspected of having PACS.

PCR were positive in thirty-three (58%) of the fifty-five patients. Twenty-three PCR were either negative or not available (See Table 2).

Among the 55 cases reported, as assessed by the doctors using the DUSOI/WONCA score of gravity, twenty-three patients showed severe (rated 3) and twenty-six showed a very severe condition (rated 4) at their first visit. Functional impairment rated scored more than 20 points (very impaired) on the six COOP/WONCA charts) by thirty-seven patients.

Comorbidity prior to PACS i.e., the distribution of the major health problems (diagnostic index) present before the acute episode are detailed in Figure 1 for the 34 first patients, coded by ICPC-2, the main classification used in primary care. This kind of distribution is usual in a family practice. It is not surprising to find a high prevalence of locomotor (L) problems in this working population in which social (Z) and psychological (P) problems are also frequent. Among nutritional and metabolic problems (T), there are seven obese patients, five of whom have had bariatric surgery and one patient is also diabetic. The circulatory (K) chapter includes a dissecting carotid aneurysm and a transient ischemic attack, both at the moment of the acute COVID-19. Each of these last two patients recovered completely from these vascular damages, which suggests that they were direct consequences of the acute COVID-19. From locomotor point of view (L), two patients have an autoimmune disease: rheumatoid arthritis, and psoriatic arthropathy. One patient had a miscarriage (W for pregnancy) during the acute COVID-19.

### 3.2. Clinical PACS Evolution on Two Years

From a clinical point of view, the evolution of the disease takes many months. After one year of follow-up, from a pragmatic point of view, and for the management by the family practice, three types of clinical evolution can be distinguished since the onset of acute COVID-19 (see Table 1 ) (Data on 52 first of the 55 patients):Grade 1, mild PACS; 16 patients (9 female, 7 male):The duration is 3 to 8 months, the impairment is mainly respiratory with fatigue, sternal pain, exhaustion with effort, no cognitive nor mnesic disorders. Other aspecific symptoms (skin redness, paresthesia, anosmia, dysgeusia, vertigo) may be present but disappear with time. These patients generally had a low DUSOI severity index and a low functional impairment score at the beginnig of the care and return to normal life after several months, sometimes with sequelae such as recurrent chest pain or taste or smell disorders.Grade 2, severe PACS; 15 patients (12 female, 3 male):The duration is 6 to 18 months, with extreme fatigue, effort exhaustion, anomia, cognitive disorder, mnesic disorder. Other symptoms of the digestive, cardiac or autonomic system could be present. Nevertheless, all the symptoms diminish after 12 to 18 months and the resumption of activities is possible with sometimes sequelae (procedural memory disorder, fatigue on exertion, sudden deep breathing). A relapse is possible. These patients most often had a high severity index (DUSOI 3 or 4) and poor functional status (COOP > 20).Grade 3, very severe PACS; 21 patients (17 female, 4 male):After 12 to 27 months (maximum at the time of writing) patients are not able to resume their activity or only part-time at the most. Exhaustion is constant, efforts impossible, cognitive revalidation useless, hypersomnia, weight gain due to inactivity, persistent severe memory disorders and of course a considerable anxiety about the future and the feeling of having contracted an unknown and incurable disease. The repercussions on the family life are very severe. The oldest patient (F, 79) is hospitalized with a hypothetical diagnosis of Alzheimer’s disease. At the beginning of the care, these patients could not be distinguished from the others and no predictive elements were found that would allow us to make a prognosis of their evolution.

#### Concerning Vaccines

Seven patients contracted COVID-19 at the very beginning of the pandemic in early 2020, before PCR tests were available. During acute COVID-19, fifty patients remained home while two patients were hospitalized. A 79 year-old-patient was hospitalized for several months, whereas another—36 years old—for two days. Eight patients had acute COVID-19 several times The relationship between PACS and vaccination is presently unclear. Seven patients were not vaccinated and all vaccinated patients were vaccinated after contracting COVID-19. Thirty-six presented a reaction to the vaccine of whom six a local reaction (pain-redness) and thirty a systemic reaction, sometimes very severe (fever for several days, fatigue, cognitive problems). Four patients temporarily improved after the vaccine while two worsened. One patient improved for 6 months before relapsing. Three patients believed that the vaccination triggered their long COVID.

### 3.3. PACS Clinical Picture

The symptoms experienced during PACS are numerous and form a recurrent picture with huge variations between patients. Previously unknown disabling state of exhaustion, inability to exert oneself with dyspnea on exertion, brain fog with memory impairment and word retrieval deficit (anomia) are the most common features. Five patients mainly had a respiratory form of PACS. Paresthesias in unexpected dermatomes, hematomas and skin spots, chest pain, involuntary movements of the limbs or fingers are also described. A short piece of verbatim of one patient is transcribed in Table 3. To give a global picture, the symptoms for each patient are aggregated and represented at the Figure 2 as a word cloud using wordclouds.com (accessed on 1 September 2022).

It is not possible to determine a date of onset for PACS. It is only possible to know when a doctor mentioned the diagnosis. Indeed, the evolution of acute COVID-19 to PACS is insidious and the patient does not always make the link between the symptoms and this new disease, which is in any case not recognised by the many doctors consulted. The identification of PACS took between 2 and 20 months after the acute COVID-19, resulting in a high degree of medical wandering and a feeling of being unrecognized or abandoned among many patients The patients who came by for a consult were previously given a wide range of assessment or diagnoses. A young girl, an elite athlete, suffering from exertional exhaustion, was called a lazy teenager by his teacher. Such diagnosis as angina pectoris, pulmonary embolism, multiple sclerosis, depression, fibromyalgia, burnout, Alzheimer, generalized anxiety disorder or post-traumatic shock were coined in the emergency department reports or by the consulting specialists.

### 3.4. Nuclear Imaging Interest

The SPECT-CT is interesting to follow the cerebral evolution. Fifteen patients (out of the 52 first) underwent a follow-up SPECT-CT after 4 to 9 months. Eight patients have an improved image showing regression of the previously described anomalies well correlated with the general improvement of their condition. Seven patients who are still very ill and severely disabled with more than 25 points on the COOP/WONCA scale have worse images with protocols such as: *“appearance of a right thalamic hypofixation and parieto-occipitotemporal hypofixation currently more marked on the left than on the right”* (not displayed), or *“with appearance of a right posterior parietal involvement compared to the previous examination”* (not displayed).

Out of the 55 patients of this series, SPECT-CT was requested in thirty-two patients, for whom all 3 of the following clinical criteria were met:Clinical symptoms suggesting a brain disorder in the context of the COVID-19 pandemicA degree of severity of 3 or 4 on the DUSOI/WONCAA functional status of more than 20 points on the COOP/WONCA

Having a positive PCR is not a determining condition to affirm a PACS. In this series one of the patients with a negative PCR did not even remember having COVID-19. PCR “proof” was not available in 23 of 55 patients, either negative or because the procedure did not exist at the beginning of 2020. On fifteen SPECT-CT requested, two of which are documented in Table 3 and Table 4 and three represented in Figure 3, Figure 4 and Figure 5, all but two showed altered cerebral perfusion, sometimes extensive. In two cases of PACS without memory impairment, there is no impairment of thalamic or subthalamic perfusion. In one reported case, it was not requested due to the patient’s desire to become pregnant. NMR are generally non-contributory except for mere minimal lesions. A follow-up NMR showed normalization of an aneurysmal lesion present in the patient during acute COVID-19.

### 3.5. Narrative and SPECT-CT Images of Some Exemplary Cases

Some of the characteristic features are underlined and highlighted here by relevant SPECT-CT images. Other cases are described in the clinical research report [39].

#### 3.5.1. Patient MGA010

This patient is forty-six years old, asthmatic, suffers from Meniere’s syndrome with severe vertigo and from common disabling migraine. The patient had never before complained about or been treated for a mental health problem. In November 2020, the patient developed acute COVID-19 with pneumonia (ground glass areas on the X-ray), and was treated at home while having the following symptoms: fatigue, dyspnoea at the slightest effort, chest pain and anosmia. After one and a half months, the patient writes this text message: “*For the past few weeks, I have been having obsessive and fixed ideas, I have experienced paranoia about my colleagues, violent anxiety and sometimes morbid ideas with a consequent state of despair, crying spells that cause chest pain, recurrent nightmares, and I sometimes have the feeling of being in a waking sleep where I am aware of having very strong chest pains, painful bones and joints …In the morning, I have no strength, no motivation, I feel reclusive and even persecuted …There is something wrong*”.

Subsequently, the chest pain persists and dizziness worsens, to the point that she has twelve dizzy spells in just two weeks. She finds it difficult to concentrate, can’t stand noise, loses her words and her memory. She is exhausted and develops sleep disorders. She was unable to work. In May 2021, the decrease in cerebral vascular flow is evident on the CT-SPECT images (See Figure 4).

The same patient, whose condition has gradually improved since, had a second scan (not displayed here) three months later. The protocol of this second scan was very reassuring: *a discreetly heterogeneous tracer fixation is observed, with clearer left frontal, left parietal and right parietal hypofixations, and the presence of periventricular hypocaptation. Compared to the previous workup, there is an improvement in cerebral fixation with a decrease in fixation heterogeneity and periventricular hypocaptation*. In February 2022, 15 months after having contracted COVID-19, the patient appears to improve.

#### 3.5.2. Patient MGA017

This patient is a warehouse clerk, aged 51, who was being treated for rheumatoid arthritis and brachialgia with cervical foraminal stenosis. In February 2020, he fell ill with a pseudo flu, which turned out to be an acute COVID-19 infection. At that time, a PCR was not performed. Several months later, he presented with a characteristic PACS with anxious depression, disabling headaches, exertional exhaustion, chest and muscle pain, paresthesia, visual disturbances, nervousness, eye burning, gastrointestinal disturbances, malaise, and above all, a major worry regarding his future.

The patient no longer feels like himself. The SARS-CoV-2 serology is negative. Consequently, there is no evidence of an occupational disease. In June 2021, the brain SPECT-CT shows the characteristics of vascular encephalopathy (Figure 5). In the spring of 2022, he was still off work and his functional state was very impaired. A claim for recognition of an accident at work was made impossible by the absence of a positive PCR test. Two years later, in February 2022, he has not recovered and the follow-up SPECT-CT shows a worsening of his condition (see Figure 5).

#### 3.5.3. Patient MGA005

A 59-year-old mother and cleaning lady, currently on disability for osteoarthritis, is not sure what happened to her. She had had COVID-19 with headaches, asthenia, muscle and joint pain as well as anosmia and agueusia, the latter two symptoms having persisted for 3 months. Now she sometimes has difficulty pronouncing certain words, so much so that she has stopped speaking. There are also problems with her writing abilities. She has always been a bit dysgraphic, but now she has lost confidence in herself and her knowledge, and she is afraid to make mistakes. She has problems with balance and concentration, occasional coughing fits, sometimes tightness of breathing and dyspnoea on exertion. Her state of exhaustion persists and prevents her from doing her work, she feels weak and anxious. She also suffers from memory loss, forgetting where she puts her keys and sometimes suffering from word retrieval deficit. The SPECT-CT shows small but significant brain damage (see Figure 6). By the end of the fourth month she had almost recovered (see Figure 6).

#### 3.5.4. Patient MGA058

Each patient has a unique life and health journey. But this one is very exemplary. 49 years old, executive managerial functions. He has already experienced a burnout, sleep disorders after a divorce and has had bariatric surgery for obesity. He was a great sportsman, bordering on sports addiction. He has done and acute COVID-19 three times. The first in March 2020, contracted at work, when there was no PCR. He stayed in bed for two weeks. The second in October 2020 with a positive PCR with fatigue and dizziness. The third in November 2021, with anosmia and dysgeusia. Interestingly the SARS COV 2 serology is negative in September 2021. But since the beginning of 2020 he has been consulting for severe cognitive and memory problems. The psychiatrist says Burnout. The neurologist says Alzheimer. A first SPECT-CT in April 2021 shows an alteration of cerebral perfusion. 18FDG PET-CT and lumbar puncture rule out Alzheimer’s. The patient was referred to the psychiatrist who prescribed Rilatine. In June 2022, 19 months after the first Covid, a second SPECT-CT confirmed the first one and the diagnosis of PACS was made. The neurological and behavioral disorders were severe. The cognitive state is very altered and fluctuates: sometimes he is as he was before, sometimes he feels lost and does not know where he is - he can no longer stay in a restaurant with friends because he is overwhelmed with information, he may completely forget why he is in a store, he may totally forget what he did during the previous hour when he has just gone for a walk with a friend. His cognitive and memory impairment has a strong impact on his social life; friends drift away and no longer understand him. He has incomprehensible physical alterations; his hands open by themselves: while he is carrying a shopping package, his hands open and the package falls. There is a loss of control and the object is dropped. He has paraesthesia of the fingertips, lateral hand tremors, sometimes very sharp, unknown before. He tried to resume sports in September 2020 but he is quickly exhausted and has to mow his lawn in several times. Since 2020 he has started to blow up his car, indicating a change in spatial perception. The imaging requested are displayed in Figure 6.

## 4. Discussion

### 4.1. Clinical Approach

Between May 2021 and July 2022, the data of fifty-five patients are exposed, of which thirty-two of high concern were imaged by SPECT-CT. Their management in family medicine is described, highlighting the importance of medically unexplained symptoms and attention to the person as well as the use of nuclear imaging in the assessment of severely affected patients.

The general context is that of the COVID-19 pandemic. These cases have been progressively highlighted in general practice consultations since May 2021 and, little by little, the notion of PACS has emerged as a coherent explanation for severely altered functional status in known patients. The very initial questioning was made during contacts with an abnormally tired patient with impaired memory who suddenly improved after two Comirnaty vaccines. This was the first case (see case MGA001 in Table 3 and in [39]). Since then, more and more patients have been and are still being identified as carrying varying degrees of PACS stigma.

In view of the polymorphous clinical picture of this syndrome, it is not surprising that emergency room colleagues or specialists have put forward diagnoses that are as diverse as they are multiple. Angina pectoris, Alzheimer’s disease, pulmonary embolism, hyperventilation, fibromyalgia, traumatic shock, burnout, anxiety attacks and post-traumatic stress syndrome have all been evoked as peremptory diagnoses that are destabilizing for the patient. Whether in the emergency room or with specialists, the cross-sectional doctor’s vision of the patient is limited to the present moment. For their part, the GP has a a long-term, repetitive, longitudinal view of the patient’s life and well-being, and can assess changes in health status over time. The initial diagnosis is essentially made with a clinical narrative approach [35], based on carefully listening to the patients on multiple occasions. The relationship with most patients had been established for many years, and it was very clear that these patients were undergoing a profound change in their condition. This study has a mixed-method approach, with both a quantitative and a qualitative component.

The symptoms presented by all patients evoke the same clinical picture made in many studies [15,25]. Most patients were not aware that their condition was related to COVID-19. The triad of exhaustion, exertional dyspnea and memory impairment seems to be recurrent. Cerebral disorders dominate, either by thalamic and subthalamic damage (memory loss), cortical damage (brain fog, anomia, hallucination, paresthesias, abnormal movements), or by bulbar damage (anosmia, dysgeusia, dizziness, orthostatic disorder), although in anosmia, both central and peripheral alterations could be demonstrated [56]. No patients have contributory NMR images, although NMR can reveal cerebral microvascular lesions in severe COVID-19 [57,58].

SPECT-CT were ordered in thirty-TWO patients judged severely affected and reporting significant functional impairment. Cerebral perfusion changes are visible in twenty-nine on thirty-two patients. The lesions found are consistent with the severity of the problem experienced by these patients. Brain SPECT-CT, as a diagnostic tool for demonstration of brain damage, measures brain perfusion and indirectly brain oxygenation. SPECT-CT was useful in 90% of suspected PACS patients showing brain disorder. This kind of result is highly unusual in a family practice setting. In daily practice, the selection bias is the rule as the doctor is searching for cases and a 29/32 (90%) efficiency is hence very surprising. 18FDG PET-CT allows brain metabolism to be directly assessed [59]. In a recent study by Verger et al., 47% of the scans of 143 patients with suspected neurological PACS were visually interpreted as abnormal [51]. 18FDG PET-CT has a superior sensitivity over SPECT-CT and provides better contrast and spatial resolution. Studies suggest superiority of 18FDG PET-CT over SPECT-CT, but the evidence base for this is actually quite limited [60].

The spectacular but very expensive 18FDG PET-CT images enforces the use of SPECT-CT as much more cost-effective from an environmental and economical point of view in primary care for the diagnosis of PACS brain perfusion disorders. Nevertheless, the question of which patients should benefit from this procedure has to be addressed cautiously.

Fifteen affected patients each had a follow-up SPECT-CT after three to nine months. The SPECT-CT showed a clear improvement in eight cases and an aggravation in seven, corresponding with the clinical improvement or aggravation of the condition. The improvement in metabolism in the PACS at six months has already been shown using 18FDG PET-CT [61]. In the patients discussed in this study, the SPECT-CT examination looks to be useful for the follow-up of the most severe cases.

### 4.2. Global Indicators of Severity and Functional Status

Our indicators of severity (DUSOI/WONCA) and functional status (COOP/WONCA) are specific to general practice and by-products of the WONCA. The DUSOI/WONCA is an indicator of the severity estimated by the doctor, while the COOP/WONCA charts is a reliable indicator [62] of the patient’s opinion of his or her condition. Many publications deal with the impact of PACS on the health status of patients, sometimes with similar indicators [63], more detailed ones [64], or for example some specific to fatigue [65]. Our indicators aim to estimate the overall condition of a patient and not the precise impact on a single function. However, patients have other intercurrent health problems and the indicators may be influenced by these new elements. It is important to note that the clinical elements, i.e., the clinical examination, the severity indicator (DUSOI) and the functional status indicator (COOP) that determine whether to request a CT-SPECT, have significant relationships with the severity of the outcome, both from the physician’s and the patient’s perspective, as shown in Table 2.

### 4.3. Limitations

Clinical medicine tries to integrate the experience of the clinician, the values of the patient and the best scientific information available. These three pillars are also the pillars of EBM [66]. But in this case, there is no—or very limited—evidence yet. This evidence is under construction as patients and doctors work together to develop knowledge. It is a type of action research [67] by which, as J. Pols points out, *it is up to us to articulate the knowledge that patients develop* [68].

This study is not without its limitations. This work was carried out in a busy family medicine practice in pandemic time and does not benefit from all the rigor that high-level research requires. In general practice, the present study is observational. Examinations are carried out if they are beneficial to the patient. A practitioner would not ask for SPECT-CT without expected benefit for the patient. In his seminal paper on problem-solving in general practice, Yan McWhinney wrote that “*the disease presented to family doctors is often in an unorganised state*” [69]. The practitioner then confronts the information provided by the patient with his or her knowledge and constructs a frame of reference acceptable to him and to the patient. Clinical decisions and methods of patient care should be based on controlled experiments and not on intuition. However, the knowledge of practitioners can also contribute to this and be studied, shared and challenged [70].

It is necessary to emphasize also that the field of research on PACS and the number of publications on this subject are growing exponentially. The initial observation of the present study dates back to May 2021 and since then the knowledge on PACS has advanced considerably. There is no pre-organised framework since the condition is new, undescribed and not part of the stock of knowledge accumulated to date. It is the doctor’s ability to listen to the patient, to be surprised and to think about a new situation that will be central to the process of uncovering the new condition. At the same time, the doctor’s experience is profoundly transformed by the research since he is an actor in it [67]. Consequently, the quality of the observations changed between the first and the fifty-fifth patient in the presented study.

The reproducibility of the SPECT-CT technique in PACS deserves to be studied, as well as its comparison with 18FDG PET-CT. SPECT-CT is not an expensive examination, invoiced at 222 € (∼250 $) to the Belgian national insurance provider. The environmental impact of labeled Technetium (Tc-99m) must also be taken into account, although Tc-99m used in medical diagnostics has a short half-life of six hours and does not remain in the body [71]. Its main advantage is that its economical and environmental cost is much lower than that of the isotopes used for 18FDG PET-CT despite the production of large quantities of highly radioactive waste during its manufacture [72]. The practitioner requesting such radio imaging should be aware of the environmental impact of the health-related activity [73].

### 4.4. Hypo-Perfusion and Hypercoagulation Looks Central to PACS Pathophysiology

Every time the severity of the case and the cognitive disorders are in the foreground, the SPECT-CT shows perfusion abnormalities (29 abnormalities on 32 ordered). These twenty-nine patients have an encephalopathy by hypo-perfusion, and the clinical pictures are so similar that one can hardy evoke another hypothesis to explain the symptoms.

Mejia et al. suggests a deleterious effect of SARS-CoV-2 infection on systemic vascular endothelial function [74]. Hohberger et al. showed an impaired capillary microcirculation in the macula and peripapillary region [75]. Fogarty et al. showed that persistent endothelial cell activation may be important in modulating the ongoing pro-coagulant effects in convalescent COVID-19 patients and thus contributes to the pathogenesis underlying the COVID-19 syndrome [76]. Hypercoagulation, vascular complications [77] and microclots formation in COVID-19 have already been identified [78]. PACS is accompanied by increased levels of antiplasmin [79], and pericytes–the multipotent parietal cells of capillaries–may play an important role in microvascular PACS alterations [80].

Aggregated platelet, microhemorrhages and ischemia appear to play a central role in neuronal injury by reducing the blood flow with concomitant reduction in oxygen and glucose in SARS-CoV-2 infected non-human primates [81]. The altered mental status may be due to encephalopathy caused by a systemic disease or encephalitis directly caused by the SARS-CoV-2 virus itself [82], without forgetting the dramatic consequences of cognitive loss on the mental health of patients. These lesions can be compared to the skin lesions that accompany COVID-19 and PACS, especially in young subjects, with a duration of 7 to 150 days [83] and up to 22 months [84]. According to Mehta et al. these lesions clinically resemble vasculopathy, with microvascular abnormalities observed with nail capillaroscopy [85]. This is an argument for making the analogy between PACS and an autoimmune damage or disorder that affects vascular endothelial surfaces. The question arises as to whether the cutaneous vascular lesions in COVID-19 are similar to the cerebral endothelial lesions inducing the perfusion disorder in PACS. Nirenberg et al. have shown the presence of fibrin thrombi occluding capillaries and endothelial swelling without vasculitis in the toe of a COVID-19 patient [84]. Furthermore, as the skin lesions heal without consequences, one can assume that the cerebrovascular lesions found will evolve in the same way.

Chest pain on exertion could also be attributable to microvascular lesions or myocardial inflammation [86]. Singh et al. show that the impaired systemic oxygen extraction observed in exhausted PACS patients is attributed primarily to reduced oxygen diffusion in the peripheral microcirculation [87]. According to an observational study by Camazon et al., coronary microvascular ischemia is the underlying mechanism of persistent chest pain [88]. The perfusion abnormalities, cognitive and memory impairment could be explained by the expression of ACE2 in the brain stem and other receptors in the cortex by the vascular wall, which makes them vulnerable to the virus [82,89].

Several studies highlight the hypo-perfusion and subsequent hypo-metabolism as one of the pathophysiological explanations of the clinical symptoms. Hyper-inflammation caused by COVID-19 may be mediated by MCA which has also been hypothesized to cause PACS symptoms [90]. The brainstem regulates respiratory, cardiovascular, gastrointestinal and neurological processes and has a relatively high expression of ACE2 receptors compared to other brain regions [91]. A perfusional disorder could then explain hyperventilation, abdominal pain or central anosmia for example. Central perfusion abnormalities, as seen in the present study, and subsequent hypometabolism could explain the cognitive and memory disorders. One can then understand why two patients in this study, who did not not present with thalamic and sub-thalamic alterations while the cortical perfusion is very disturbed, leading to intense fatigue, brain fog and effort exhaustion but, surprisingly, not to any memory disorder.

However, the mechanism of action of the SARS-CoV-2 virus on the brain is subject to many hypotheses including viral persistence [11,81,92]. Galan et al. have shown that individuals with PACS showed *significantly increased levels of functional memory cells with high antiviral cytotoxic activity* implying that Sars-CoV-2 infection is still ongoing in PACS patients [93]. This viral persistence could explain the recurrence of symptoms described by the patients.

### 4.5. Impact of Imaging Diagnosis on Patients’ Experiences

In most cases of PACS, patients know they are sick, but no one can put a name to their illness and they go from doctor to doctor, being identified as medical wanderers with their nameless disease [94]. Knowing with certainty that the patients’ experiences are not fantasised but correspond to clear and visible lesions was in each case a shock for the twenty-nine patients whose SPECT-CT showed altered cerebral perfusion. Despite the anxiety that comes with the diagnosis, the patients are reassured to know that they “*are not crazy*”, that they “*knew that there was something*”, and that “*their family, their employer will finally believe them*”. The clinician’s gradual knowledge increase and understanding of the clinical picture will reassure the patients who have not had SPECT-CT.

For example, a young girl, a passionate gymnast who was called a lazy teenager by her teacher, can finally explain why she could no longer make any effort and why it took her more than 6 months to be at 80% of her capacity again. A truck driver, who had to stop driving to sleep during his working hours, was reassured by the letter of explanation sent to the occupational physician. Another patient now knows from the SPECT-CT images that the previously unknown condition she experienced for more than six months is not a mental disorder nor an Alzheimer.

### 4.6. Severe Reactions to Vaccines in Many Patients

All but seven patients were vaccinated. Most were vaccinated after contracting COVID-19. Thirty patients have had significant systemic effect side effects lasting from one day to two weeks. None of them is willing to be vaccinated again as they are afraid of late effects. Some people believe that the vaccine and not COVID-19 is responsible for the onset, aggravation or reactivation of their disease. Three had fleeting improvement or up to six months of their symptoms before a recurrence. Although SARS-Cov-2 vaccination is not associated with a decrease in quality of life or worsening of symptoms [95] and that there is no strong evidence to suggest that vaccination improves symptoms of PASC [96], this remains a sensitive issue because individual experiences are more important to patients than anything that science can demonstrate.

### 4.7. An Empathetic Therapeutic Approach

The announcement of the diagnosis is in itself a therapeutic act and must be carefully considered. There are as yet no definitive, evidence-based recommendations for the management of PACS. Patients should be managed pragmatically and symptomatically [97]. Psychotherapeutic care, neurocognitive revalidation and physiotherapy seem to provide some comfort [98,99], although the efficacy of these treatments are difficult to assess. The concept of neuroplasticity, known in other cerebral pathologies [100], can be used here to encourage the patient to revalidate his memory. In the Netherlands, a two-arm multicentre randomised controlled trial (RCT) showed that a comprehensive revalidation program called “Fit after COVID” significantly reduces fatigue [101]. Whether through cognitive exercises, or through the patient’s use of cognitive applications for smartphones, patients must be encouraged to slowly and gradually regain lost ground. One could of course wish for neurocognitive revalidation by specialists in the field, but revalidation centers are lacking in Belgium. Getting the body back into action, despite breathing difficulties and pain, is absolutely essential while respecting the limits of each patient. Physiotherapy also has an important role to play [102]. Recently, the Belgian social security system has facilitated the access of patients identified as PACS to several health professions such as psychologists, speech therapists, dieticians, occupational therapists and physiotherapists [103].

To date, no drug has been shown to be of proven value in the treatment of PACS, although many substances are proposed as symptomatic treatments [75,92,104,105,106,107,108]. Considering the known coagulation disorder in COVID-19 [109] and the low vascular perfusion seen in PACS, one could propose administering with due caution aspirin at a low dose to limit the risk of micro-thrombi at the level of the damaged vessels as is done in COVID-19 in high doses [110,111].

### 4.8. Further Studies

It is currently not possible to distinguish between vaccine immunity and natural immunity in patients. Moreover, there is no biological proof that these cases can all, without exception, be tagged as PACS. In an attempt to disentangle those questions, a collaboration was initiated with the Department of Microbiology of the Catholic University of Leuven (Rega Institute; https://rega.kuleuven.be/) (accessed on 1 September 2022) and the Karolinska institute (Petter Brodin; https://ki.se) (accessed on 1 September 2022). In the framework of the European Consortium for Genetic and Immunological Studies on COVID-19 (COVID-HGE consortium; https://www.Covidhge.com) (accessed on 1 September 2022), the patients in this study will benefit freely from extensive blood tests. This joint genetic and immunological study tries to understand the occurrence of PACS in some of the COVID-19 patients [12,46,112]. To date, forty-eight patients have agreed to donate their blood for these in-depth analyses. The full results will only be available in a few months and will be published in a separate article.

## 5. Conclusions

PACS syndrome has a high prevalence in primary care for those who want to see it.Clinical skills and narrative medicine are essential to identify and understand patients’ experiences. This needs time, open mindedness and empathy.Cerebral hypo-perfusion demonstrated by SPECT-CT seems to correlate with the clinical symptoms in a cohort of PACS patients. This needs further studies.Uncertainty about the primary acute infection is a problem. The participation of 48 patients to the European Consortium for Genetic and Immunological Studies on COVID-19 will probably provide some answers and further questions.The impact of PACS is substantial, with many social and economic implications.

## Figures and Tables

**Figure 1 viruses-14-02000-f001:**
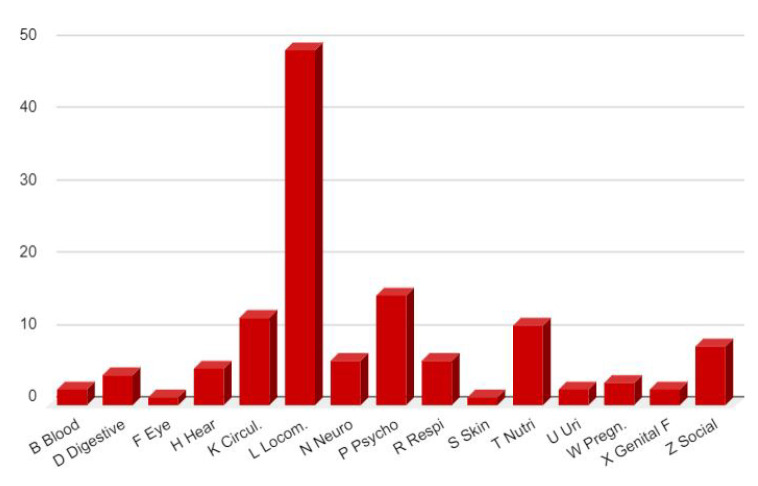
Number of health problems or diagnosis found in the diagnostic index of the first 34 patients at the time of diagnosis of PACS, coded in ICPC-2 chapters (excluding PACS related symptoms and diagnosis).

**Figure 2 viruses-14-02000-f002:**
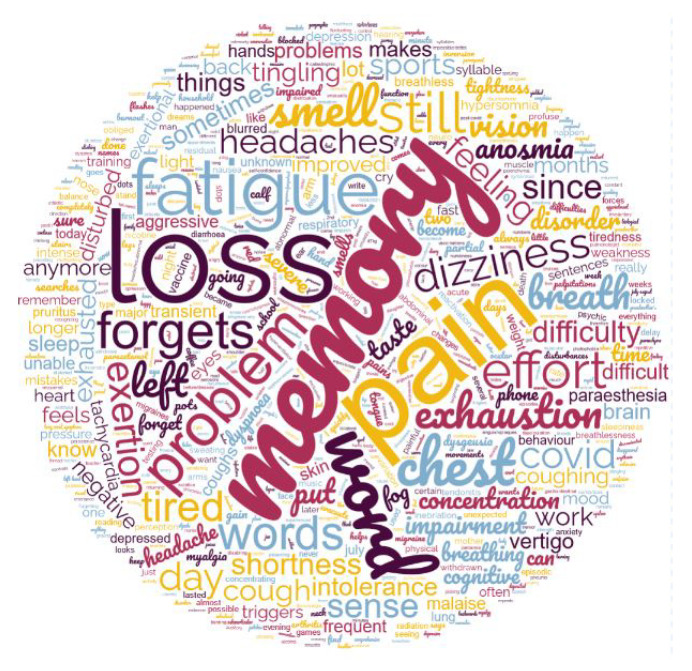
Patients express recurrent complaints. Here we have included all the complaints transcribed in English in the observation recordings of the first 34 patients seen in the family medical practice. This terminological view, edited in a word cloud (wordclouds.com), gives a conceptual representation of the clinical picture of PACS recorded by doctors. The size of the words gives an idea of their recurrence. We can see immediately that loss and pain are the most important complaints, loss of memory but also loss of the word and then the innumerable symptoms characteristic of the Long covid.

**Figure 3 viruses-14-02000-f003:**
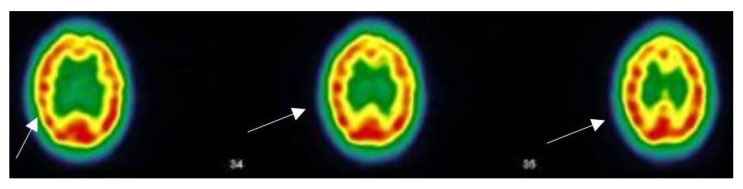
Case MGA010, F, 46, May 2021; SPECT-CT examination (ECD Tc-99m): The image shows three brain sections of the same patient.The arrows show the defects of perfusion. The red areas indicate a good tracer fixation and thus a good perfusion. The decrease in flux intensity is highlighted by the color change from red to yellow (see arrows). This indicate hypofixation and thus ischemia. Protocol; “*Heterogeneous tracer fixation with clearer left frontal, left parietal and right parietal hypofixations. No preservation of the sensory-motor cortices. The fixation in front of the grey nuclei and the cerebellum is correct. Presence of periventricular hypocaptation. Conclusion: Evidence of heterogeneous tracer fixation and periventricular hypocaptation compatible with vascular-type cerebral damage.*” (Images and protocol: Drs Bouazza and Mahy, Vesalius Hospital, ISPPC, Belgium).

**Figure 4 viruses-14-02000-f004:**
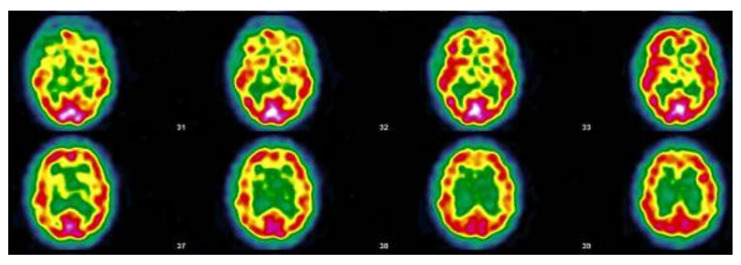
Patient MGA017; SPEC-CT (ECD Tc-99m); The image shows eight brain sections of the same patient. The red areas indicate a good tracer fixation and thus a good perfusion. Yellow patches appearing in red areas indicate hypofixation and thus ischemia. Protocol: “*Heterogeneous tracer fixation with bilateral temporal, bilateral frontal, left posterior parietal, right parieto-occipito-temporal hypofixation. Discrete preservation of the sensory-motor cortices. The fixation in front of the grey nuclei is correct. Right cerebellar hypofixation. Cortico-subcortical atrophy with periventricular hypocaptation as an indirect sign. Conclusion: Scintigraphic examination compatible with a cerebral pathology of vascular type. Moderate cortico-subcortical atrophy.*” (Images and protocol; Drs Bouazza & Mahy, Vesale hospital, ISPPC, Belgium).

**Figure 5 viruses-14-02000-f005:**
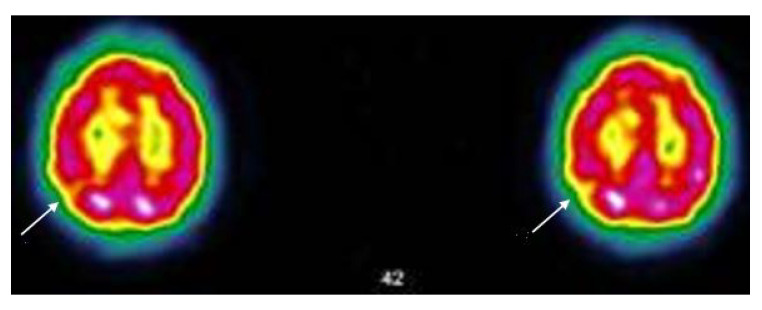
Case MGA005; SPECT-CT (ECD Tc-99m) of 25/05/21; The image shows two brain sections of the same patient. The red areas indicate a good tracer fixation and thus a good perfusion. Yellow patches (see arrows) appearing in red areas indicate hypofixation and thus ischemia Protocol; “*heterogeneous tracer fixation with clearer hypofixations left frontal, left parietal, right parietal. No preservation of the sensory-motor cortices. The fixation in front of the grey nuclei and the cerebellum is correct. Presence of periventricular hypocaptation. Conclusion: Evidence of heterogeneous tracer fixation and periventricular hypocaptation compatible with vascular-type cerebral damage.*” (Images and protocol; Drs Bouazza & Mahy, Vésale Hospital, ISPPC, Belgium).

**Figure 6 viruses-14-02000-f006:**
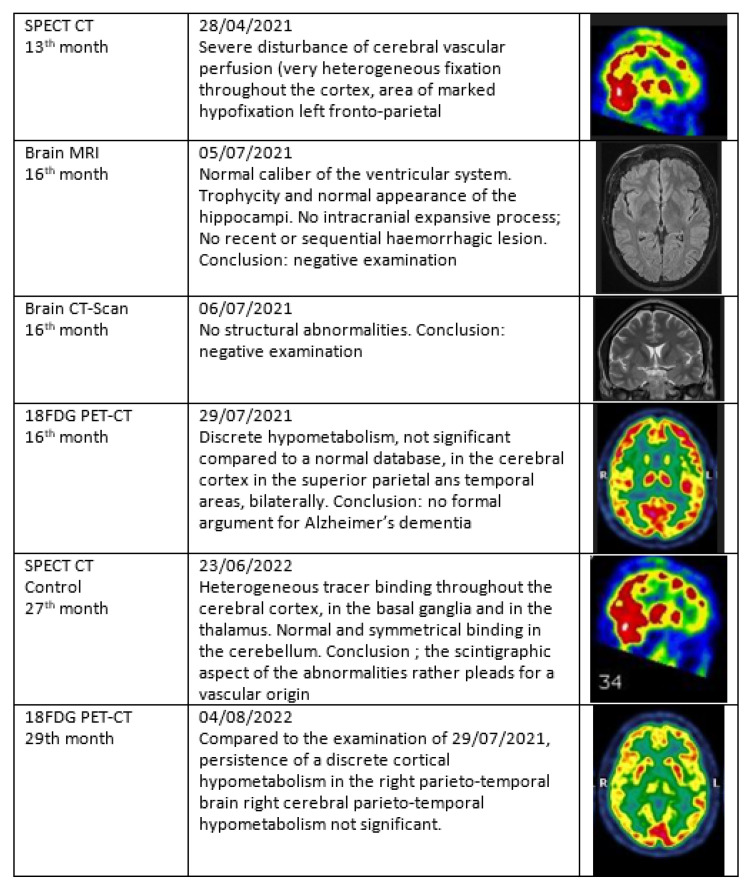
Case MGA 058, M, 49, senior executive. Acute COVID-19 on 1 March 2020. Persisting cognitive loss and exhausting fatigue, still off work after 27 months. Nuclear imaging shows its value in the follow-up of patients with PACS. Brain MRI and CT do not contribute in this case. First and second SPECT-CT highlights the vascular flow problem while 18FDG PET-CT provides information on the metabolism of the brain, although not considered significant here. (Images; courtesy of Dr Fabienne Richelle, St Luc Clinic, Bouge, Belgium).

**Table 1 viruses-14-02000-t001:** Evolution over time of a cohort of patients seen in family medicine with PACS after an acute COVID-19 episode (Centre médical Janson, Charleroi, Belgium, 2021–2022).

Status	Recovered		Still Sick
	**Mild**	**Severe**	**Very Severe**
Grade	1	2	3
Label	mild Long Covid	severe Long Covid	very severe Long Covid
Length	3 à 8 months	6 à 18 months	> 18 months
Number	16 patients (9 f, 7 m)	17 patients ( 13 f, 4 m)	22 patients (18 f, 4 m)
Symptoms	No after-effects	Minor sequelae (e.g. difficult exertion or minor loss of working memory)	Major fatigue, exertional exhaustion, difficulty concentrating, emotional disturbances, paresthesia, persistent memory problems
Capacity	Normal course of life resumed		Unable to resume normal life

**Table 2 viruses-14-02000-t002:** Main characteristics describing a cohort of 55 clinically PACS patients who presented themselves in a family practice in Belgium, during 2021–2022. The characteristics of two groups; group 1: cured mild and severe long COVID versus group 2: still ill very severe long COVID, (see Table 1) were compared to each other using the appropriate statistical tests. Ordering of first SPECT-CT, gravity indicator (DUSOI) and functional status indicator (COOP) which are all clinical considerations seem to have significant relationships with the severity of the outcome. Second SPECT-CT is ordered only to still sick patients. Months after acute COVID are calculated at the last patient assessment, i.e., the month before the submitted data collection. Both show a relationship with the outcome, as expected.

		Outcome					
		Recovered Mild and Severe		Still ill Very Severe			
		**n**	**%**	**n**	**%**	**Test**	* **p** *
Sex	Female	22	55	18	45	1.528 ^#^	0.216
	Male	11	73.3	4	26.7		
Number of COVID episodes	1	28	63.6	16	36.4	3.918 *	0.117
	2 or 3	4	44.4	5	55.6		
# of vaccines	0	6	85.7	1	14.3	4.292 *	0.202
	1	0	0	1	100		
	2	12	66.7	6	33.3		
	3	15	51.7	14	48.3		
Vaccine reaction	Yes	18	50	18	50	4.238 *	0.121
	No	9	75	3	25		
	Not vaccinated	6	85.7	1	14.3		
Vaccine reaction type	Local	4	66.7	2	33.3	0.800 *	0.658
	Systemic	14	46.7	16	53.3		
First SPECT	No	19	86.4	3	13.6	10.617 ^#^	0.001
	Yes	13	41.9	18	58.1		
First SPECT status	Yes	13	43.3	17	56.7	0.034 *	1.000
	No	1	50.0	1	50.0		
Second SPECT	No	28	73.7	10	26.3	12.766 ^#^	**<0.001**
	Yes	3	20.0	12	80.0		
Second SPECT status	Improved	3	37.5	5	62.5	3.281 *	0.123
	Worsened	0	0	7	100		
PCR test in the first episode of COVID	No	15	65.2	8	34.8	0.448 ^#^	0.503
	Yes	18	56.3	14	43.8		
DUSOI	2	6	100	0	0	13.847 *	**0.001**
	3	18	78.3	5	21.7		
	4	9	34.6	17	65.4		
		**Mean**	**SD**	**Mean**	**SD**		
Age (years)		42.9	15.6	42.0	12.9	0.222 ^##^	0.825
COOP Total score		20.8	3.7	23.4	2.3	2.758 ^##^	**0.008**
Months after acute COVID		13.3	8.9	18.3	5.9	2.347 **	**0.019**

# Chi-square test, * Fisher’s exact test, ** Mann-Whitney U test, ## Independent samples *t*-test.

**Table 3 viruses-14-02000-t003:** Acute COVID-19 and PACS experience of patient MGA001 interviewed by GK. The patient is a Kurdish speaker expressing herself in French. The verbatim, translated in English, has been slightly improved.

The acute COVID-19 experience	*Before I got sick with COVID-19, I was generally feeling fine because I was used to my condition. But when I got COVID-19, everything became more difficult …“ ”Before having COVID I was mentally in a pretty good place, a little depressed but not much. But with COVID, I was at my lowest point. A week to ten days later, I became extremely sick and more and more afraid. I could hardly see, I couldn’t even hold my phone. I felt like I had no oxygen left in my brain and I had to go to the window to breathe. I had awful headaches. I was thinking, ‘I’m going to die, I’m going to die,’ ‘Why me? Why did I get COVID?’. I wanted to die. Life seemed very dark to me, and I didn’t feel like living, I didn’t feel like eating. Even tea, which I usually like to drink, disgusted me. (…) For more than three weeks, I didn’t eat, I lost weight, I couldn’t sleep, I woke up at night, I cried and cried …I didn’t know how to do anything.*
The PACS period	*For nine months I didn’t laugh, I was always tired, I didn’t go out, I was always in a chair. I had to eat all the time, I gained 7 kgs. I would have hunger attacks, and when I didn’t eat, I would shake. In the morning, when I woke up I ate, at night I woke up and ate. Every morning I was waiting for the night to come and every night I waited for the morning. The days were endless because I was sick, I did nothing. I couldn’t stand the TV or the noise. Before COVID, I thought I was a beautiful woman but after when I looked at myself in the mirror, I said to myself ‘I am so old, as if I had aged ten years’. I forgot a lot, words, names, …I had to repeat to myself ‘I must not forget, I must not forget’. My brain was working backwards. I was angry for no reason. I was wondering ‘when will I die?’*

**Table 4 viruses-14-02000-t004:** Patients MGA001 & MGA013 with clinical PACS. Age, sex, date of acute COVID-19, date of suspicion of PACS diagnosis, date of first SPECT-CT, EMRs notes, SPECT-CT protocols (Courtesy of Drs Bouazza and Mahy, Vesalius Hospital, ISPPC, Belgium).

Acute Symptoms	Long-Lasting Symptoms	SPECT-CT Protocol
MGA001, F, 48		
13 October 2020 Throat pain, rhinorrhea, bad aches, severe fatigue and headache, but no breathing difficulties, dysgeusia, anosmia. Stays at home, cured after 12 days. Home care only	11 November 2020 Pain in both eyes, ocular pruritus, rapid ocular fatigue, noise intolerance, memory loss (forgets to pick up her daughter at school), concentration problems, remains isolated in her room, dyspnea at the slightest effort and at speech, almost continuous osteoarticular and muscular pains often with headaches, abnormal dreams, depressive feeling, fatigue, post-exertional malaise (PEM)	27 July 2021 “*On the images taken, left fronto-parietal, left frontal and left thalamic hypofixation is observed. No preservation of the sensory motor cortices. The fixation in front of the cerebellum is correct. Conclusion: Scintigraphic examination compatible with a cerebral pathology of the vascular type with clearer left fronto-parietal, left frontal and left thalamic vascular disorders*”.
MGA013, F, 39		
3 March 2021 Cough, aching, elevated temperature, headache, 20 days in bed, loss of taste, loss of smell, severe tinnitus, 20 days in total. Home care	5 October 2021 Hearing loss in right ear, balance always disturbed, dizziness, loss of vision, quickly tired, severe weight gain, quickly out of breath, became depressed, pain in left hip every night, post-vaccinations headaches, memory loss, word retrieval deficit, repeats herself and doesn’t realize it, forgets which groceries she went to get, has trouble concentrating, disseminated myalgia, insomnia, loss of sense of direction	17 November 2021 “*Heterogeneous tracer distribution throughout the cortex, with more marked hypofixation in the bilateral predominantly left superior parietal, left parietal, bilateral medial temporal and bilateral predominantly right parieto-occipital areas. Diffuse subcortical periventricular hypofixation. The basal ganglia and cerebellum show preserved and symmetrical tracer uptake. Scintigraphic image suggestive of vascular damage in the broad sense.*”

## Data Availability

De-identified patient records are available on-line for research purposes.

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
