# Peer review of "Follow-Up of a Cohort of Patients with Post-Acute COVID-19 Syndrome in a Belgian Family Practice"

_viruses, 2022, doi:10.3390/v14092000_

Round 1

Reviewer 1 Report (New Reviewer)

The author conducted a study on patients with PACS in Belgian family practice. Fifty-two patients who suffered from COVID-19 and were very ill were recruited followed by a qualitative and quantitative description of their health status. It was found that PACS is substantial with many social and economic implications in the Belgian population and primary care plus narrative and patient view is essential in dealing with PACS syndrome. Moreover, SPECT-CT is able to assess severely affected patients, especially those with neurological damages. PACS is a big concern nowadays and its causes, diagnosis, and treatment remain unknown. This study will help us understand PACS symptoms and its physio-pathological features. However, some concerns jeopardize the significance and quality of this work. A major revision is required before acceptance of the manuscript for publication.

 Major concerns:

1.     The authors did not meet the requirements of the “Article” type. In the manuscript, case studies were included. In addition, the manuscript does not follow the format required by the Journal in general. There are many subtitles in the “Discussion” section.

2.     20 patients do not have PCR tests and leaving uncertainty of some results. It is not clear whether the conclusion is affected by this. Clarification of results is required.

3.     There is no statistics at all in this study. The results/findings and conclusion are not convinced.

4.     The results are not well presented, interpreted, and thus hard to understand. As a result, there is no focus in the discussion. The discussion section should be condensed and simplified and focus on the main points from this study, instead of being diverse and optimistic.

5.     The post-acute COVID-19 findings were not justified by age, sex, and other confounding factors. For example, women seem to be more susceptible to PACS in this study. Age seems another factor influencing the occurrence of PACS as individuals with COVID-19 infection in the age of 25 – 64 are more likely to develop PACS in the study. 

Minor concerns:

1.     Table 1, no legend. Should be presented more professional, such as titles in the first row, Mean time of illness (months): 15 (2-27), etc. what about Nr. Of Man?  What laboratory proof? Give a percentage of vaccination in your studied subjects. The same thing for PCRs, SPECT-CT and DUSOI, COOP charts, etc.

2. Tables 2 and 3 can be Supplement tables.

3. In Figure 2, no full names were provided.

4.     Figure 3 is hard to interpret. It is strongly suggested to use a table instead of a figure.

5.     Line 318 -344, Grades can be presented in a table

6.     4.1 Aim of this study should be put at the end of the” introduction” section

7.     5. Take-home message, should be a conclusion

Author Response

Reviewer 2 Report (New Reviewer)

The manuscript by Jamoulle et al offers an interesting view of post-acute COVID-19 syndrome (PACS) in a small clinical study conducted in Belgium. The authors highlight a list of symptoms associated with PACS and collected from 52 patients from whom some underwent brain imaging to detect brain abnormalities. This manuscript provides critical awareness on the onset of PACS, pathology, treatments, and the socio-economic impact in the lives of people that suffer from this disease. Importantly, the authors demonstrate the spectrum of PACS duration, although in a very small cohort. The manuscript is very well written, a bit long but enjoyable to read. I will only suggest the authors to comment on whether the use of antivirals or monoclonal antibodies and reduction of the viral course will perhaps make a difference on the onset of this syndrome. How long after having COVID-19, did the patients developed PACS. Did they have COVID-19 by PCR with no symptoms?

Minor suggestions are indicated below:

Line 77: “fee for service group practice of two family doctors, serving about 2.000 patients” is this two thousand patients? It reads as only two patients.

Figure 4 and 5: arrows to depict the defect in perfusion are very confusing. It is not clear what are they pointing at.

Line 279: “He has done the Covid three times” Is this referring to the test or is this referring to the disease. The authors should clarify.

Round 2

Reviewer 1 Report (New Reviewer)

The authors have addressed the comments and made the necessary changes.  

This manuscript is a resubmission of an earlier submission. The following is a list of the peer review reports and author responses from that submission.

Round 1

Reviewer 1 Report

The article by Jamoulle et al. investigated the impact of Long Covid among 34 patients who have been experiencing extreme fatigue, effort exhaustion, brain fog, anomia, memory disorder, anosmia, dysgeusia, and other multi-systemic health problems for about 2 years (2020 to 2022) in a family practice setting in Charleroi, Belgium. A variety of clinical signatures were monitored. The Long Covid patients suffered from memory loss and pain in most cases. Additionally, many patients could not work or were able to work part time only. The authors selected 15 patients for Brain single-photon emission computed tomography (SPECT-CT) for fifteen patients with a high severity index and a functional status highly impaired. The data showed that fourteen of fifteen patients had vascular encephalopathy, which supports the hypothesis of a persistent cerebral vascular flow disorder among the patients with Long Covid. It is an interesting manuscript and probably many will be appearing in the coming days detailing the impact of Long Covid. Overall, it is great study and shows a connection between Long Covid and the psychological impact of CIOVID-19 on the general population. While the cohort is limited, it gives a good picture of Long Covid. My only comment is that the Introduction section is too long. It can be shortened by just focusing on Long Covid. It is also interesting to know that the WHO has not come up with a name for Long Covid and the authors had to rely on social media nomenclature.

Reviewer 2 Report

I have read with interest the case series by Jamoulle and colleagues. Indeed, defining long COVID is important for a variety of reasons, including the basic clinical factors for correct diagnosis and disease management. Several cases series have been published so far, but more research is needed to understand this COVID-19-attributable disease.

However, the paper here proposed need thorough revision before being considered for publication. Overall, it is not clear what authors are pointing out: they wanted to give some evidence from family medicine management of long COVID patients, but then added data on CT exams (clearly not a technology available in family medicine settings, except maybe for certain ones).

Again, the structure of the paper needs to be revised by an expert researcher: the text should be restructured as a scientific paper both as writing and meaning. Some examples here:

  • The use of Covid instead of COVID-19 is incorrect for a scientific paper
  • The sentence “In a large number of previously infected patients, which, depending on the authors, can vary from 10% to more than 50%, Covid is no longer just an acute syndrome.” Does not take into account the differences in epidemiological data do not “depend on the authors”, but on other aspects, namely study population and setting, study design and timing, case assessment, etc…
  • Methods are not clear: the section is just list of “tools” without. The possibility to understand what actually authors did
  • Statistical analysis is missing
  • Results are presented in a confusing manner

For these and other reasons, it is very hard to continue with the reviewing of this paper.